# Development and evaluation of a Physiotherapy-led, WHO-ICOPE-Based, Person-Centered Integrated Care Program (PTICOPE) module to enhance intrinsic capacity in older adults: Protocol for a randomized controlled trial

**Nurhazrina Noordin[1,2], Akkradate Siriphorn[2], Yu Chye Wah[3], Maria Justine[1,2]***

**1** Centre for Physiotherapy Studies, Faculty of Health Sciences, Universiti Teknologi MARA Selangor, Puncak Alam Campus, Puncak Alam, Selangor, Malaysia, **2** Department of Physical Therapy, Faculty of Allied Health Sciences, Chulalongkorn University, Bangkok, Thailand, **3** School of Physiotherapy, Faculty of Allied Health Professions, AIMST University, Kedah, Malaysia

* maria205@uitm.edu.my

## Abstract

### Background

Declining intrinsic capacity (IC), encompassing domains such as locomotion, cognitive function, vitality, vision, hearing, and psychological well-being, is prevalent among older adults, impacting independence and quality of life (QoL). This study aims to develop and evaluate the Physiotherapy-led Person-Centered Integrated Care for Older People (PTICOPE) based on the WHO-ICOPE framework to enhance IC among older adults in Malaysia.

### Methods

This is a 12-week, multicenter, randomized controlled trial involving 70 community-dwelling older adults aged 60–75, recruited from three *Pusat Aktiviti Warga Emas* (PAWE) (Activity Center for Older Adults) in the Northern region of Malaysia. Participants will be randomized to either the intervention group, receiving the PTICOPE module workbook and guided use, or the control group, receiving general IC information, healthcare education, and self-care management. The recruitment of participants for this study has not yet commenced. Recruitment is expected to start after completing the validation of the PTICOPE module, however, it is anticipated that the recruitment start date is in February 2025 and will end in August 2025. Primary outcomes, including locomotor, psychological, cognitive, vitality, visual, and hearing functions using validated scales, will be collected at baseline, 4th, 8th, and 12th week of the study period. Secondary outcomes will evaluate QoL, activities of daily living, urogenital health, and oral health at baseline and 12th week. The normality of data will be checked. The independent t-tests, Chi-square tests, paired

**Data availability statement:** No datasets were generated or analysed during the current study. All relevant data from this study will be made available upon study completion.

**Funding:** The author(s) received no specific funding for this work.

**Competing interests:** The authors have declared that no competing interests exist.

t-tests, and Repeated measures ANOVA will be used for data analysis, with a significant level at $p < 0.05$.

## Discussion

This study will develop a PTICOPE based on the WHO-ICOPE framework and test its efficacy in the older population residing in the community. The findings of this study hold the potential to establish an evidence-based approach for enhancing IC among community-dwelling older adults.

## Trials registration

Thai Clinical Trials (Number: TCTR20241029007).

## Introduction

Intrinsic capacity (IC) is a novel concept referring to the combination of an individual's physical and mental abilities that define healthy and active aging, especially in later life [1,2]. The World Health Organization emphasizes that the well-being of older adults must be holistic, encompassing optimal functional capacity across six domains: locomotion, cognitive function, vitality, vision, hearing, and psychological function, which serve as biomarkers of healthy aging [3]. Older adults can achieve a higher quality of life (QoL) in their later years when they are in a supportive environment and reach the peak of each health domain, thereby reducing societal burdens [2]. However, as age advances, maintaining optimal IC becomes challenging due to natural aging processes, comorbidities, and healthcare services disruption [4,5].

Numerous studies have reported a high prevalence of IC decline, ranging from 49.9% to 84.5% [6–8]. A recent systematic review and meta-analysis indicated a pooled prevalence of IC decline of 67.8% within community settings (95% CI: 57.0-78.5%; p < 0.001) in which China had the lowest prevalence at 66.0% (95% CI: 53.2-78.9%), compared to 73.0% (95% CI: 59.8-86.3%) in other countries [1]. Another review and meta-analysis, using the WHO's Integrated Care for Older People (ICOPE) screening tool, identified an overall IC impairment prevalence of 55.0%, with domain-specific prevalence rates as follows: locomotion (17.5%), cognition (18.2%), psychological well-being (12.1%), vitality (8.5%), vision (17.9%), and hearing (14.4%) [9]. Notably, the domains most and least affected by decline appear to differ by country. For instance, locomotion was reported as the most impacted domain in China, with psychological function least affected [10], while in India, locomotion was most impacted and vitality the least [8]. These variations may suggest the need for setting-specific, person-centered approaches and multidisciplinary care strategies to support older adults in the community considering differences in the sociodemographic factors, cultural norms, and local healthcare policy and infrastructure.

In response to the increasing global challenge of IC decline, the WHO launched the Integrated Care for Older People (ICOPE) framework in 2017, also known as WHO-ICOPE intending to provide a systematic approach to meeting the diverse health needs of older adults and to promote healthy aging within the community settings [11]. The framework consists mainly of the guidelines for a person-centered assessment process that guides the creation of individualized care plans informed by screening outcomes, followed by regular monitoring to assess effectiveness. The framework also has an additional emphasis on the value of involving community members and caregivers and encourages fostering connections with the broader environment and society. Several studies have applied the WHO-ICOPE framework to

implement a person-centered care model for older adults. For instance, while the AMICOPE model provides a foundation, it has notable limitations, particularly its minimal focus on sensory domains [12]. Another approach, the INSPIRE-ICOPE program, was developed for older adults in affluent countries with generally higher education levels [13], which could reduce its applicability in culturally diverse settings such as Malaysia. Additionally, one study that used the ICOPE guidelines concentrated on a limited set of domains, specifically addressing cognitive, locomotor, and psychological aspects [14]. Given these limitations, further research is necessary to establish a more comprehensive, person-centered model that spans all IC domains, incorporating physical activity, personalized guidance, and routine follow-ups. Consequently, there is a need for healthcare providers to create robust, person-centered integrated care models that foster high functional capacity and improved well-being.

It is evident from previous studies that strategies for enhancing IC varied, which could be due to the researchers' aim to target the specific needs of the study population. Due to that, interventions to enhance IC and functional abilities should be individualized, focusing on specific goals set by each individual, and can be delivered by any member of the multidisciplinary team. Furthermore, as Malaysia's older population, particularly those in rural communities, may have limited access to comprehensive healthcare services, a person-centered approach that adapts and adopts the ICOPE framework could provide tailored interventions.

Findings indicate that locomotion is the most commonly affected domain [9], and that IC decline is independently associated with increased risks of frailty, disability, falls, fractures, and immobility [10], suggesting that physiotherapy plays a critical role in maintaining and enhancing IC among older adults. Therefore, we aimed to develop a Physiotherapy-led Person-Centered Integrated Care for Older People (PTICOPE) program and evaluate its effectiveness in improving IC among older adults living in the community. We hypothesized that older adults participating in the PTICOPE program would experience significant improvements in their IC domain scores, including mobility, vitality, psychological function, cognitive function, hearing, and vision, compared with the control group. This community-based program is expected to empower older adults to achieve greater independence, enhance their functional abilities, and improve their overall well-being. Additionally, it provides guidelines for physiotherapists and other healthcare professionals working with community-dwelling older adults to regularly screen IC and monitor outcomes effectively.

## Methods

### Study design

This study is a multicenter, randomized, parallel-group approach with a 1:1 allocation, randomly assigning participants to either the intervention or control group. The study is consistent with the guidelines of Standard Protocol Items: Recommendation for Trials (SPIRIT) (Fig 1). The protocol follows the CONSORT flowchart, outlining participant progression through the randomized controlled trial, with follow-ups scheduled at Weeks 4, 8, and 12 following the initial baseline assessment (Fig 2). Participants are informed of their assigned group. The recruitment of participants for this study has not yet commenced. Recruitment is expected to start after completing the validation of the PTICOPE module, however, it is anticipated that the recruitment start date is in February 2025 and will end in August 2025.

### Study settings

The study will be conducted at three community-based centers for older adults, also known as *Pusat Aktiviti Warga Emas* (PAWE), specifically in the districts of Sg Petani, Yan, and Kulim, Kedah, Malaysia. These centers were selected as they provide basic facilities that

| | STUDY PERIOD | | | | | | | |
|---|---|---|---|---|---|---|---|---|
| | Enrolment | Allocation | Post-allocation | | | | | Close-out |
| TIMEPOINT** | -$t_1$ | 0 | $t_1$ Baseline | $t_2$ Week 4 | $t_3$ Week 8 | $t_4$ Week 12 Close-out | *etc.* | $t_x$ |
| **ENROLMENT:** | | | | | | | | |
| **Eligibility screen** | X | | | | | | | |
| **Informed consent** | X | | | | | | | |
| *[List other procedures]* | | | | | | | | |
| **Allocation** | | X | | | | | | |
| **INTERVENTIONS:** | | | | | | | | |
| *Intervention Group* | | | X | X | X | X | - | - |
| *Control Group* | | | X | - | - | - | - | - |
| | | | | | | | | |
| **ASSESSMENTS:** | | | | | | | | |
| *Baseline Measures* | | | X | | | | | |
| *IC domains (primary outcomes)* | | | X | X | X | X | -. | - |
| *Secondary outcomes* | | | X | - | - | X | -. | - |

 *Recommended content can be displayed using various schematic formats. See SPIRIT 2013 Explanation and Elaboration for examples from protocols.
 **List specific timepoints in this row.

**Fig 1. SPRIT schedule for enrollment.**

allow sufficient space for physical and social activities for community-dwelling older adults. Furthermore, these centers have a large number of registered older adults, around 200 to 300 members for each center, who have similar socioeconomic and cultural backgrounds.

## Participants

Community-dwelling older adults registered with PAWE will be invited to participate in the study. Most PAWE members are independent in activities of daily living and have controlled comorbidities. The inclusion criteria for participants are as follows: (1) aged 60 to 75 years; (2) able to walk independently or with partial assistance from an assistive device; (3) manageable comorbidities, as evidenced by their ability to participate in PAWE activities; (4) able to understand and communicate in either Bahasa Malaysia or English; and (5) willing to participate in the study. The exclusion criteria are: (1) uncontrolled comorbidities, such as hypertension, type 2 diabetes mellitus, cardiovascular disorders, or severe joint conditions; (2) Dementia or Alzheimer's disease; and (3) blindness or deafness.

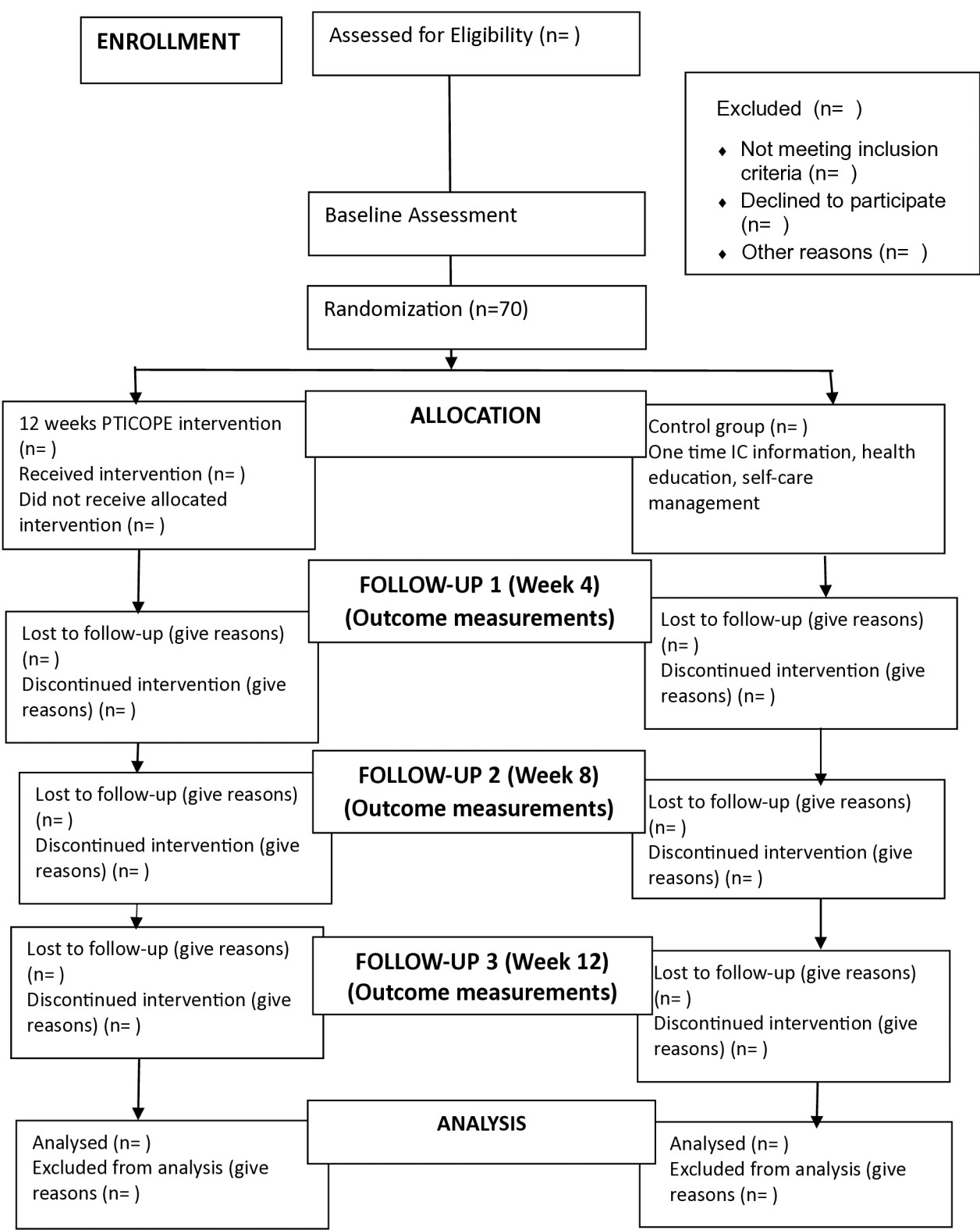

**Fig 2. The CONSORT flow diagram.**

## Sample size determination

The sample size was calculated using G* Power 3 software for a repeated measures ANOVA (Baseline, week 4, week 8, week 12 measurements), aiming for a statistical power of 90%, an alpha level of 0.05, an estimated medium effect size (0.25), and a standard deviation of 2.5 for the primary outcomes and a correlation among repeated measures of 0.5. The selection of these parameters was guided by a previous study examining multicomponent exercise interventions in older adults to enhance, which reported a meaningful change of 1 point in the Short Physical Performance Battery (SPPB) as a clinically significant threshold [14].

Based on the initial analysis, a minimum of 60 participants is needed to detect a significant effect within each group. To account for potential attrition, such as participant dropouts due to withdrawal, health issues, or other factors, about 15 to 20% increase was applied. This adjustment results in a target sample size of 70 participants, divided equally into two groups (35 per group) to ensure the study maintains sufficient power to detect meaningful differences in IC changes between the intervention and control groups.

## Randomization and allocation concealment

Participants will be randomly assigned to intervention and control groups using computer-generated randomization (http://www.randomizer.org/). A research assistant will handle the preparation of the envelopes by placing a carbon paper on top of the allocation paper, inserting both into a foil wrapper, and sealing them inside an envelope. Each envelope will be signed across the seal for authenticity. A total of 70 envelopes will be prepared with 35 allocated for the intervention group and 35 for the control group. Each envelope will be manually labelled as either "intervention" (1) or "control" (0). Once prepared, all 70 envelopes will be thoroughly shuffled to ensure randomness. A unique number sequentially from 1 to 70 will then be marked on the front of each envelope using a pen. The carbon paper inside will transfer this number onto the allocation paper. Finally, the envelopes will be placed in a plastic container in numerical order and ready to be used for the group allocation. After recruiting a participant, the research assistant will sequentially open an envelope, with the number inside indicating the participant's group assignment. In this study, neither the participants nor the research assistants will be blinded to group assignments; however, the data collector will be blinded to group assignments to minimize measurement bias.

## Intervention

**Development of PTICOPE module.** The development process of the PTICOPE module includes two main steps: (1) creating the PTICOPE module based on the WHO-ICOPE guidelines and evidence gathered, and (2) validating the module through content validation (with experts) and face validation (with older adult participants). This approach is designed to enhance IC in older adults through a tailored, person-centered framework. Initially, data from literature and systematic reviews are gathered to identify factors and interventions related to IC decline. We also adopt the construct of the Health Belief Model (HBM) to guide the development of the module as shown in Table 1. The HBM recognizes and addresses the social context in which health behaviours take place that can be incorporated into the PTICOPE intervention to increase knowledge of health challenges, enhance perceptions of personal risk, encourage actions to reduce or eliminate the risk, and, eventually enhance the sense of self-efficacy to continue changing [15,16].

The contents of the PTICOPE workbook will be tailored to the Malaysian context by aligning with the intervention and management suggested by WHO-ICOPE (Table 2), featuring contents that are educational, culturally relevant, practical exercises with self-tracking tools to support

**Table 1. The application of the health belief model in the PTICOPE module.**

| HBM Constructs | Features of PTICOPE |
|---|---|
| Perceived Susceptibility and Severity | Participants will be educated about the importance of intrinsic capacity (IC) and how its decline can impact activities of daily living, independence, well-being, and quality of life. The participants will be informed of their assessment findings to help them recognize their vulnerability to IC decline. |
| Perceived Benefits | The participants will be informed about the benefits of the PTICOPE intervention in enhancing the IC domains, not only through the exercises and healthy diet but also the indirect effects of seeking personalized care from other health-care disciplines. In the long run, these practices can prevent disability, promote independence, and support active aging. |
| Perceived Barriers | Through ongoing monitoring of the PTICOPE intervention via WhatsApp and face-to-face meetings, the researchers and participants will address the potential barriers that limit their adherence to exercise and dietary advice. Through discussion between the researchers and participants, guidance and ways can be suggested to overcome the barriers. |
| Cues to Action | The researchers will remind the participants regularly to continue achieving and revising the goals that they have agreed to at the beginning of the intervention. |
| Self-Efficacy | The elements of the PTICOPE module include strategies for managing health and enhancing IC through goal-setting, and feedback from the researchers. The researchers will use techniques such as demonstration and positive reinforcement during the weekly meeting and communication through WhatsApp. Positive reinforcements from the researchers may build the confidence of the participants to continue following the recommended activities in the PTICOPE module. |

older adults' intrinsic capacity and ease of understanding. The input from experts, among other include physiotricians, geriatricians, psychologists, occupational therapists, dieticians, and input from practitioners and educationists in the validation process ensure collaborative efforts from several healthcare disciplines that can address all aspects of aged-care. The workbook is designed with interactive features like visual aids and graphic instructions to enhance participants' engagement, making the workbook an effective and user-friendly tool for older adults.

Moreover, the contents of the module are also aligned with WHO's 13 ICOPE recommendations, which are customized to meet the specific needs of older adults in the respective settings [12]. Table 2 outlines the proposed contents of the PTICOPE module.

**Validation of the PTICOPE module.** The validation of PTICOPE includes a thorough content validation phase involving 12 multidisciplinary healthcare experts, representing fields such as occupational therapy, physiotherapy, geriatrics, family medicine, public health, nutrition, dietetics, audiology, psychology, and academia. Each expert receives an invitation along with an information sheet and consent form. Upon agreeing to participate, they are provided with the PTICOPE module (workbook) and the Patient Education Materials Assessment Tool for Printable Materials (PEMAT-P), a 26-item form for evaluating printed educational materials [17]. Experts will have at least two weeks to submit their feedback, after which researchers will revise the module. This process will continue iteratively until no further changes are required.

Following content validation, the finalized PTICOPE module will undergo face validation with a sample of community-dwelling older adult participants. Ten individuals from the target population, who will not participate in the study, will review the module for clarity, relevance, and ease of use. These participants will receive an invitation, including an information sheet and consent form. A group meeting will be held in a meeting room at one of the PAWE centers, where researchers will present the PTICOPE module, focusing on participants' understanding, visual appeal, and acceptability of the intervention tasks. Two research assistants will document feedback during the session, which will be audio-recorded for accuracy. This feedback will inform final refinements to the workbook, preparing it for testing.

**The intervention group.** Participants in the intervention group will undergo the PTICOPE intervention for 12 weeks (Table 2). Meetings will be held once a week for the first 4 weeks. During the initial meeting, participants will receive guidance on how to use

**Table 2. Contents of the PTICOPE module.**

| Intrinsic Capacity Domain | Activities/Interventions | Advice | Goal Setting |
|---|---|---|---|
| Locomotor | • Balance exercises (e.g., tai chi, single-leg stands) three times weekly.<br>• Strength exercises using resistance bands.<br>• Flexibility exercises like stretching or yoga. | Encourage daily low-impact physical activities like walking at moderate intensity.<br>Recommend supportive footwear and safety modifications (e.g., handrails). | Weekly mobility goals tailored to ability levels, such as aiming for 10-15 minutes of walking daily, gradually increasing time and intensity. |
| Vitality | • Nutritional guidance with a focus on protein intake, hydration, and key vitamins<br>• Monthly meal planning assistance.<br>• Weekly hydration reminders. | Recommend balanced meals with lean protein, vegetables, and hydration goal (e.g., 8 glasses daily).<br>Provide strategies for appetite stimulation if needed. | Daily dietary and hydration goals; track nutritional intake using a simple chart or apps. |
| Psychological | • Mindfulness practices (e.g., breathing exercises, meditation).<br>• Stress-reduction techniques such as guided imagery.<br>• Weekly community engagement activities (e.g., group walks). | Recommend social interactions, like weekly calls or outings with family/friends.<br>Encourage hobbies that provide cognitive engagement (e.g., gardening, reading). | Set personal well-being goals such as attending a social event weekly or completing a relaxation activity daily. |
| Cognitive | • Memory exercises (e.g., word games, memory matching).<br>• Problem-solving tasks (e.g., puzzles, strategy games).<br>• Use of cognitive apps like Lumosity for 10-15 minutes daily. | Recommend engagement in cognitive activities (e.g., puzzles) for mental stimulation.<br>Encourage reading or learning a new skill. | Set weekly cognitive goals (e.g., complete a puzzle or read a chapter daily). |
| Hearing | • Whisper test or audiometry screening annually.<br>• Environmental adjustments (e.g., using hearing aids, minimizing background noise). | Advise on ear care (e.g., avoid loud environments, use ear protection if needed).<br>Encourage social support for improved communication. | Aim to maintain current auditory health or improve understanding with assistive devices.<br>Set quarterly hearing check-ins. |
| Vision | • Vision exercises to reduce eye strain.<br>• Regular eye exams (annually).<br>• Home environment adjustments (e.g., improve lighting). | Recommend eye protection (e.g., sunglasses outdoors).<br>Suggest routine visual health monitoring, such as checking for blurriness. | Set visual health goals (e.g., reduce screen time, follow up on eye examinations). |

the workbook, be informed of their baseline measurement outcomes to help strategize their care pathway, set personal goals, learn the exercises outlined in the workbook, and have the opportunity to ask any questions about the intervention. Participants are encouraged to contact the researchers via WhatsApp at any time throughout the week for further clarification regarding the intervention.

In the following 3 weekly meetings, to be held at the same locations, researchers will check that the workbook is being used correctly, ensure that exercises are performed according to the established goals, and address any additional concerns. After these four meetings, the first follow-up measurement will be conducted, with participants informed of their results to assist them in revising their personal goals as needed. The second and third follow-ups will continue as scheduled, with participants reminded of their appointments at the centers for further outcome measurements.

**The control group.** Participants in the control group will receive a one-time session on IC information, including health education on the importance of exercise, a healthy diet, and self-care management. The control group will not receive the PTICOPE booklet. During the first meeting, the participants in the control group will be informed of a series of monitoring and measurement of outcomes (weeks 4, 8, and 12). A WhatsApp reminder will be sent to their mobile phone to inform them regarding the timing of the follow-up

## Outcome measures

**Primary outcomes.** The repeated measures of the six domains of IC (cognitive decline, limited mobility [locomotor], malnutrition [Vitality], visual impairment, hearing loss,

and depressive symptoms) are the primary outcomes of this study. These outcomes will be assessed at baseline and weeks 4, 8, and 12, using the WHO ICOPE screening tool. The ICOPE screening tool is a sensitive instrument to detect IC among community-dwelling older adults that does not require a substantial workforce [18]. It is used as the first step in each care pathway that covers all six core domains of IC to identify declines in IC across the six domains [19]. The psychometric properties of the ICOPE screening tools to assess IC domains among older persons from different settings have been reported in a few studies [20–22]. However, a systematic review found that the methods for assessing each of the IC domains differ substantially across studies and there is no consensus on the best method to compute an eventual global compound score to evaluate IC [23]. On the other hand, further improvements are needed in the vitality dimension of the ICOPE screening tool to enhance its sensitivity in identifying individuals at risk of malnutrition [24]. Hence, for this study, some measures of outcomes for the IC domains will be based on those validated in the Malaysian population.

We will use the validated Malay version of the Mini-Mental State Examination (MMSE) to measure cognitive function [25]. This tool includes items on orientation (5 points), registration (3 points), attention and calculation (5 points), recall (3 points), and language (9 points), with a maximum score of 30. A higher MMSE score indicates better cognitive function.

To assess locomotor or mobility function, we will use the Short Physical Performance Battery (SPPB) [26,27]. The SPPB comprises three components: gait speed (measured over 4 meters), lower limb strength (five-repetition chair rise), and static balance (side-by-side, semi-tandem, and tandem stands), with each component contributing up to 4 points. The total SPPB score ranges from 0 to 12, with higher scores indicating better locomotion.

For vitality assessment in the ICOPE screening tool, we will use questions on unintended weight loss of more than 3 kg in the past three months and appetite loss. This study will employ the validated Malay version of the short-form Mini Nutritional Assessment (MNA) [28]. The MNA has a maximum score of 14, with higher scores indicating better nutritional status.

For hearing and vision assessments, we will follow previously reported procedures, using self-reported hearing and vision impairment [29]. Participants will be categorized as having total or severe loss (0 points), moderate loss (0.5 points), or normal to mild loss (1 point).

Psychological function, as indicated by depression level, will be assessed using the validated Malay short-form Geriatric Depression Scale (GDS-15) [30]. This scale consists of 15 items with "Yes" (1 point) and "No" (0 point) responses; a higher score indicates a greater level of depression.

**Secondary outcomes.** For the secondary outcomes, we will gather data on instrumental activities of daily living (IADL) and quality of life (QoL) to gain further insights into the participants' overall well-being throughout the intervention. These secondary outcomes will be measured at both baseline and 12 weeks following the intervention. IADL will be assessed using the Lawton-Brody Instrumental Activities of Daily Living Scale, which evaluates independence in various functions like managing finances, taking medications, and handling transportation (Lawton & Brody, 1969).

To assess quality of life, the EQ-5D-5L questionnaire will be administered, a widely used tool for evaluating health-related QoL [31]. This tool has two parts: a descriptive system where participants rate aspects such as mobility, self-care, usual activities, pain/discomfort, and anxiety/depression on a scale from 'no problems' to 'extreme problems.' The second part, the EQ Visual Analogue Scale (EQ VAS), allows participants to indicate their general health on a vertical scale, ranging from 'The best health you can imagine' to 'The worst health you can imagine.

## Data collection and monitoring

Data collection and monitoring of participants' progress will be conducted by trained research assistants who are physiotherapy students or qualified physiotherapists. While the delivery of the intervention will be done by the main researchers (First and third author) assisted by the research assistants. Before participant recruitment, to ensure consistency across different settings, the research assistants will undergo thorough training, including understanding the protocol of the PTICOPE and hands-on practice with all outcomes measures for measuring the IC, guiding participants through tailored exercises, and providing education on self-management strategies. The research assistants will also monitor progress, adjust interventions to individual needs with guidance from the main researchers.

Recruitment of study participants will be conducted at three PAWE after obtaining research ethics approval and permission from the person in charge of the centers. The researchers will invite all participants who are members of the PAWE via letter of invitation, flyers, and verbal invitations. Those who volunteer to participate will be invited to a briefing session to inform the purpose and procedure of the study. Participants who provided informed consent and fulfilled the inclusion criteria will undergo the 12-week RCT. All participants will undergo an evaluation of the sociodemographic data, and primary and secondary outcomes at baseline. Evaluation of primary outcomes will be repeated at week 4, week 8, and week 12. Evaluation of secondary outcomes will be repeated at week 12. Once baseline data has been collected, participants will be randomly assigned to either the intervention or control group. The intervention group will receive the PTICOPE module in the form of a workbook and include comprehensive guidance on its use to support IC. In contrast, the control group will receive general information on the importance of maintaining IC, along with health education on the importance of exercise, and healthy diet, and self-care management which include recommendations to seek relevant healthcare services based on their baseline IC measurements.

## Data analysis

All statistical analyses will be conducted using SPSS version 29, with a significance level set at $p < 0.05$. Data will first undergo cleaning and analysis in SPSS, where frequency distributions and graphical representations will be used to identify outliers and missing values. For normality testing of interval data, the Kolmogorov-Smirnov and Shapiro-Wilk tests will be applied. Descriptive analysis will be conducted for participants' demographics and baseline characteristics to describe data using tables, averages, standard deviations, ranges, frequencies, and percentages. Where appropriate, statistical comparisons between the intervention and control groups will be conducted, utilizing the independent t-test for continuous data and the Chi-square test for categorical data.

To evaluate the intervention's impact on the primary outcomes, a repeated measures ANOVA will be conducted after data collection, as the primary outcomes involve measurements at baseline, week 4, week 8 and week 12, to evaluate the significant effects of time, intervention, and their interaction. For the secondary outcomes which involve measurements at baseline and week 12, the paired-t-test will be performed to determine the impact of the intervention.

The non-parametric tests will be performed if the main outcomes are not normally distributed that include the Friedman test and Wilcoxon signed-ranked test for the primary and secondary outcomes, respectively.

In the event of missing data due to participant dropout or incomplete data collection, an intention-to-treat (ITT) analysis will be employed to maintain the validity of the randomization process. Missing data will be managed using appropriate imputation techniques,

depending on its nature and extent. For continuous variables, multiple imputation methods will be applied, while for categorical variables, methods such as last observation carried forward (LOCF) may be used. Sensitivity analyses will also be performed to assess the robustness of the results to the assumptions about the missing data.

To ensure data privacy and confidentiality, hard-copy forms collected from participants will be stored securely. All datasets will be digitized and saved on password-protected laptops, with backup copies maintained on password-protected USB drives and external hard drives. Access to these datasets is restricted to the principal researcher.

## Ethics and dissemination

Ethical approval was obtained from the UiTM Research Ethics Committee (REC/09/2024 (PG/MR/437)), and this study is registered as a clinical trial in the Thai Clinical Trial Registry TCTR20241029007) (https://www.thaiclinicaltrials.org/show/TCTR20241029007). Informed consent will be obtained from all participants, ensuring their privacy and right to withdraw at any time.

This study will adhere to ethical standards throughout all procedures. During participant recruitment, trained research assistants will meet with potential participants to explain the study's purpose, procedures, benefits, potential risks, and participants' right to withdraw at any time without consequences. Participants will have sufficient time to discuss any questions before signing the consent form. All data collected will be stored centrally, kept confidential, and anonymized. Data analysis will be conducted by the research team and used solely for study purposes within the research team.

Study findings will be submitted for publication in a peer-reviewed journal and presented at a relevant conference, such as the Asian Conference for Frailty and Sarcopenia, held annually in the region. The findings will support the proposal of a new physiotherapy approach based on PTICOPE for community-dwelling older adults.

## Discussion

The purpose of this study is to develop and evaluate the PTICOPE module, a Physiotherapy-led, WHO-ICOPE-based, person-centered integrated care program specifically tailored for older adults in Malaysia. This module aims to enhance intrinsic capacity across six key domains, namely, locomotor, vitality, psychological, cognitive, vision, and hearing, addressing unique cultural and healthcare access challenges faced by Malaysian older adults. By incorporating comprehensive exercises, goal-setting, advice, and regular monitoring, the PTICOPE module seeks to empower older adults to maintain and improve functional abilities and overall well-being within a community-based setting, thereby contributing to improved quality of life and reduced healthcare burden among the aging population.

The current PTICOPE module which is led by physiotherapists may be different from those that have been published such as the AMICOPE [12], INSPIRE ICOPE-CARE [13], and VIVIFRAIL [14], that were conducted in Spain, France, and tertiary hospitals in Spain, respectively. These differences in terms of language, socioeconomic factors, diet, lifestyles, and healthcare policy highlight the need to design specific interventions for integrated care to older adults. As far as we know, this is the first study using the constructs of the HBM to develop the features of the PTICOPE module. The applicability of the HBM constructs has been evident in several studies involving older persons with different conditions [32,33]. Nevertheless, the effectiveness of this application in the current study is yet to be ascertained once the study is completed. We also anticipate several challenges in the application of the HBM in the older population due to variations in their physical and psychological well-being.

It is also pertinent that the development of intervention to enhance IC is acceptable by the local community as one template may not apply to all due to the diverse world population of aging. This is evidenced by IC studies reported in several countries. For instance, a study reported in China and Japan revealed that IC decline is associated with social fragility and not modifiable factors [34]. In contrast, another study in China found poor self-rated health, multiple chronic conditions, recent falls, frequent physical pain, recent outpatient visits, hospitalizations, and pessimistic life expectancy were highly associated with IC decline [35]. While, a study conducted in India found age, gender, and smoking were related to IC decline [36]. These findings indicate that factors contributing to IC decline may differ across countries probably due to diversity in socioeconomic status, cultural norms, and healthcare policy, hence, may require different approaches in the personalized and integrated care pathways.

The development of the PTICOPE ensures a comprehensive review from expert panels who have at least 5 years of working experience with older adults, ensuring a strong understanding of the needs of older adults. Furthermore, older adults are also involved in the development of the module, in the face validation stage, to ensure that the module is acceptable to the study participants and effective in enhancing the well-being of the study participants.

## Strengths of the study

We would like to highlight several strengths of this proposed study. First, we will ensure that the development of the PTICOPE module involves expert review from all disciplines involved in or relevant to the care of older adults, including experienced physiotherapists, occupational therapists, and geriatricians, among others. The module's development is based on the WHO framework and current literature, followed by content and face validation to ensure it effectively covers all aspects of IC.

## Limitations and considerations

Several limitations should be noted in this study. First, this study will be conducted in a single state in the northern region of Malaysia, where the predominant members of PAWE are of Malay ethnicity. This limitation may restrict the generalization of the findings to other states in Malaysia. However, given Malaysia's diversity in culture, ethnicity, and healthcare infrastructure, this limitation is unavoidable. We suggest that each state may need to adapt the proposed module for local suitability.

The second anticipated limitation could be related to participant adherence to the module's activities. Older adults may experience challenges such as hearing or vision impairments that could impact their understanding and adherence to the study protocol. To address this, we will contact each participant weekly to remind them of their roles and set goals as outlined in the module.

Thirdly, although we have calculated the sample size based on a recommended formula, older adults may still have the potential to withdraw from the study due to multiple declines in IC domains, as noted in the literature. We hope that weekly mobile phone reminders will help reduce the attrition rate and encourage participants to stay motivated in achieving their goals and caring for themselves.

Fourthly, there are possible bias in self-reported answers, the study will use trusted and proven tools, keep participants' identities private, and ask questions in a neutral way to encourage honest responses. Combining this data with other sources like interviews or observations will make the results more reliable, and using simple, clear language will help avoid confusion.

Fifthly, there may be a long-term challenge in ensuring participants compliance to the interventions and data collection. To mitigate this potential issue, the researchers will keep in touch with participants through updates and reminders via phone calls and WhatsApp mode. The participants will be ensured of flexible schedules, simple processes, and appreciation for encouraging and motivating their involvement.

The participants fidelity and acceptability of the intervention will be ensured by regularly asking the understanding of the participants towards the PTICOPE module, regularly tracking their completed task, and gathering feedback during the follow-up sessions. Participants will be informed of their performance and goal-achievement that should be consistent with their exercise recording table during follow-ups to continue motivating them and strive for a better results and satisfaction.

## Future research directions

Future studies should involve multiple states in Malaysia to encompass the country's diverse population and culture, enhancing the generalizability of the PTICOPE module. Further research is also recommended to extend this approach longitudinally, allowing assessment of its sustainability, long-term acceptance among participants, and adherence to newly introduced self-care practices. Additionally, similar modules could be tested in different settings, such as government-funded primary care clinics.

## Conclusions

The newly developed PTICOPE module, delivered by physiotherapists with multidisciplinary input, is anticipated to have great potential for enhancing the intrinsic capacity of older adults living in the community. The PTICOPE's uniqueness and culturally tailored features may empower a person-centered approach, enabling older adults to better manage their physical and psychological well-being, resulting in an improved quality of life, greater independence, and reduced caregiver and healthcare system burdens. For future work, we plan to adapt the PTICOPE module for remote or digital formats to expand its reach and accessibility, especially for underserved or rural communities.

## Supporting information

**S1 File. Trial registration.**
(PDF)

**S2 File. Study protocol.**
(PDF)

**S3 File. SPIRIT fillable checklist.**
(PDF)

## Author contributions

**Conceptualization:** Nurhazrina Noordin, Akkradate Siriphorn, Yu Chye Wah, Maria Justine.

**Data curation:** Nurhazrina Noordin.

**Formal analysis:** Nurhazrina Noordin, Maria Justine.

**Investigation:** Nurhazrina Noordin, Yu Chye Wah, Maria Justine.

**Methodology:** Nurhazrina Noordin, Akkradate Siriphorn, Maria Justine.

**Project administration:** Yu Chye Wah, Maria Justine.

**Resources:** Yu Chye Wah, Maria Justine.

**Supervision:** Yu Chye Wah, Maria Justine.

**Validation:** Akkradate Siriphorn, Maria Justine.

**Visualization:** Akkradate Siriphorn, Maria Justine.

**Writing – original draft:** Nurhazrina Noordin.

**Writing – review & editing:** Akkradate Siriphorn, Maria Justine.

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
