## [Decision Letter · Decision Letter 0]

20 Dec 2024

PONE-D-24-52840Development and evaluation of a Physiotherapy-led, WHO-ICOPE-Based, Person-Centered Integrated Care Program (PTICOPE) Module to enhance intrinsic capacity in older adults: Protocol for a randomized controlled trialPLOS ONE

Dear Dr. JUSTINE,

Thank you for submitting your manuscript to PLOS ONE. After careful consideration, we feel that it has merit but does not fully meet PLOS ONE’s publication criteria as it currently stands. Therefore, we invite you to submit a revised version of the manuscript that addresses the points raised during the review process.

Dear Authors,The manuscript has received an overall positive feedback from the reviewers, however, some necessary changes are suggested by them to improve the overall readability as well as significance of the conveyed results. Please make the modifications as suggested by both the reviewers and resubmit for further consideration.==============================

We look forward to receiving your revised manuscript.

Kind regards,

Yogesh Kumar Jain, MPH

Academic Editor

PLOS ONE

Journal Requirements:

2. We note that your Data Availability Statement is currently as follows: “No datasets were generated or analysed during the current study. All relevant data from this study will be made available upon study completion.”

Please confirm at this time whether or not your submission contains all raw data required to replicate the results of your study. Authors must share the “minimal data set” for their submission. PLOS defines the minimal data set to consist of the data required to replicate all study findings reported in the article, as well as related metadata and methods (https://journals.plos.org/plosone/s/data-availability#loc-minimal-data-set-definition ).

If your submission does not contain these data, please either upload them as Supporting Information files or deposit them to a stable, public repository and provide us with the relevant URLs, DOIs, or accession numbers. For a list of recommended repositories, please see https://journals.plos.org/plosone/s/recommended-repositories .

3. We note that the original protocol that you have uploaded as a Supporting Information file contains an institutional logo. As this logo is likely copyrighted, we ask that you please remove it from this file and upload an updated version upon resubmission.

Reviewers' comments:

Reviewer's Responses to Questions

**Comments to the Author**

1. Does the manuscript provide a valid rationale for the proposed study, with clearly identified and justified research questions?

Reviewer #1: Yes

Reviewer #2: Yes

2. Is the protocol technically sound and planned in a manner that will lead to a meaningful outcome and allow testing the stated hypotheses?

Reviewer #1: Yes

Reviewer #2: Yes

3. Is the methodology feasible and described in sufficient detail to allow the work to be replicable?

Reviewer #1: No

Reviewer #2: Yes

4. Have the authors described where all data underlying the findings will be made available when the study is complete?

Reviewer #1: No

Reviewer #2: Yes

5. Is the manuscript presented in an intelligible fashion and written in standard English?

Reviewer #1: Yes

Reviewer #2: Yes

6. Review Comments to the Author

You may also provide optional suggestions and comments to authors that they might find helpful in planning their study.

Reviewer #1: The authors present the framework for their proposed study of the PITCOPE module. They appropriately describe the question of interest, the planned PITCOPE procedure, how participants will be recruited, and what metrics will be assessed when on each participant.

What is lacking a bit is more detailed information for the sample size calculations. Why was a paired t-test used (when the primary outcome mentions 4 time points, so a repeated measures ANOVA might be more appropriate). Additionally, what is the estimated means/standard deviations or effect size used for the calculation of the needed sample size? These are important values to include for reproducibility purposes. It is also important to include what material were used to inform the values used for the means/st devs and/or the effect size.

The randomization method also needs attention. As written, it is very unclear how the value of 0 or 1 that gets put into the envelope will be determined. It seems that a sheet with 70 0's and 70 1's (70 sets of numbers) will be created. From there, envelopes will only have one of these numbers. It a simple coin flip method is used (simple randomization), it is unclear how equal sample sizes in each treatment group will be achieved. A more detailed description of the exact method of randomization is needed, and the method used should mention how the end result will be an equal number of subjects in each intervention arm.

For the data analysis section, the authors mention that the data will be checked for normality with appropriate tests. However, there is no mention as to what will happen if the data is found to be not normally distributed. Will data transformations be performed? Will alternate methods be used (Wilcoxon signed rank)? These items should be mentioned in the text as potential alternatives.

Reviewer #2: The manuscript presents a well-designed protocol for a randomized controlled trial to evaluate the efficacy of a Physiotherapy-led Person-Centered Integrated Care for Older People (PTICOPE) module in enhancing intrinsic capacity among older adults in Malaysia. Here are the key points addressing the review questions:

1. The manuscript provides a valid rationale with clearly identified research questions. The study aims to address the prevalent issue of declining intrinsic capacity in older adults, which impacts their independence and quality of life.

2. The protocol is technically sound and planned to lead to meaningful outcomes. It employs a 12-week, multicenter, randomized controlled trial design with clear intervention and control groups, comprehensive outcome measures, and appropriate statistical analysis methods.

3. The methodology is feasible and described in sufficient detail to allow replication. The protocol outlines participant recruitment, randomization, intervention details, outcome measures, and data analysis plans.

4. The authors have stated that all relevant data from the study will be made available upon study completion.

5. The manuscript is presented in an intelligible fashion and written in standard English, with a logical structure and clear language.

Comments for the authors:

• Consider providing more details on the PTICOPE module workbook content and how it will be developed based on the WHO-ICOPE framework.

• Clarify the specific roles of physiotherapists in delivering the intervention and how they will be trained to ensure consistency across centers.

• Discuss potential limitations of the study and how they will be addressed, such as potential bias in self-reported measures or challenges in long-term follow-up.

• Consider including a plan for process evaluation to assess the implementation fidelity and acceptability of the PTICOPE module.

• Provide more information on how the sample size of 70 participants was determined and whether it is sufficient to detect clinically meaningful differences between groups.

Overall, the protocol is well-designed and has the potential to contribute valuable insights into enhancing intrinsic capacity among older adults. Addressing these minor points will further strengthen the manuscript.

7. PLOS authors have the option to publish the peer review history of their article (what does this mean? ). If published, this will include your full peer review and any attached files.

**Do you want your identity to be public for this peer review?** For information about this choice, including consent withdrawal, please see our Privacy Policy .

Reviewer #1: No

Reviewer #2: **Yes: ** RICHARD AMOAKO

---

## [Author Response · Author response to Decision Letter 1]

7 Jan 2025

Dear Professor Yogesh,

(Academic Editor)

Thank you for your constructive feedback to our manuscript entitled "Development and evaluation of a Physiotherapy-led, WHO-ICOPE-Based, Person-Centered Integrated Care Program (PTICOPE) Module to enhance intrinsic capacity in older adults: Protocol for a randomized controlled trial" to PLOS ONE Journal.

We are glad to receive insightful feedback from you and the reviewers who had spent time and effort to provide the comments for improvement. Thus, it is with great pleasure that we resubmit our article for further consideration. We have incorporated changes that reflect the detailed suggestions you have provided. We also hope that our edits and the responses we provide below satisfactorily address all the issues and concerns you and the reviewers have noted.

To facilitate your review of our revisions, we have provided the point-by-point response to the questions and comments delivered in your mail dated last 20th December 2024. Please see the followings:

i. Table 1: Responses to the Editorial

ii. Table 2: Responses to Reviewer 1

iii. Table 3: Responses to Reviewer 2

All amendment/changes are indicated in tracked changes, in red colours, as shown in the manuscript.

Once again, we thank you for the opportunity to make the amendment based on the valuable comments from the reviewers. We have worked hard to incorporate your feedback and hope that these revisions persuade you to consider accepting our manuscript for publication.

Best Regards,

Maria Justine

(Corresponding Author)

UiTM Puncak Alam Campus

Centre for Physiotherapy Studies, Faculty of Health Sciences, Universiti Teknologi MARA ,Selangor, Puncak Alam Campus, Puncak Alam, Selangor Malaysia

Email: maria205@uitm.edu.my

Table 1. Responses to the Editorial

Editorial/Journal Requirements Authors’ Responses

1. Please ensure that your manuscript meets PLOS ONE's style requirements, including those for file naming. The PLOS ONE style templates can be found at the link provided in mail.

We have named all necessary file according to the PLOS ONE’s requirements.

2. We note that your Data Availability Statement is currently as follows: “No datasets were generated or analysed during the current study. All relevant data from this study will be made available upon study completion.” Please confirm at this time whether or not your submission contains all raw data required to replicate the results of your study. Authors must share the “minimal data set” for their submission. PLOS defines the minimal data set to consist of the data required to replicate all study findings reported in the article, as well as related metadata and methods.

- Authors do not need to submit their entire data set if only a portion of the data was used

in the reported study.

- If your submission does not contain these data, please either upload them as Supporting

Information files or deposit them to a stable, public repository and provide us with the

relevant URLs, DOIs, or accession numbers. For a list of recommended.

Thank you for highlighting this.

As for now we do not have any data yet, as data collection has not started.

We have written the following in the submission system:

Deidentified research data will be made publicly available when the study is completed and published.

3. We note that the original protocol that you have uploaded as a Supporting Information

file contains an institutional logo. As this logo is likely copyrighted, we ask that you

please remove it from this file and upload an updated version upon resubmission

We have removed the institutional logo and resubmit the study protocol without the logo.

Table 2. Responses to Reviewer 1

NO COMMENTS RESPONSE

1 The authors present the framework for their proposed study of the PITCOPE module. They appropriately describe the question of interest, the planned PITCOPE procedure, how participants will be recruited, and what metrics will be assessed when on each participant. Thank you for the thoughts. We hope and really appreciate your valuable inputs to improving the manuscript and hence, be accepted for publication.

We have amended and improved the manuscript to the best of our capacity. The point-by-point amendments followed subsequently.

2 What is lacking a bit is more detailed information for the sample size calculations. Additionally, what is the estimated means/standard deviations or effect size used for the calculation of the needed sample size? These are important values to include for reproducibility purposes. It is also important to include what material were used to inform the values used for the means/st devs and/or the effect size.

• Thank you for your valuable suggestions to relook the sample size calculation.

• Consistent with another review, we have provided more details on the sample size calculation by mentioning estimated effect size and other parameters.

• See, Sample size determination – 1st paragraph.

3 Why was a paired t-test used (when the primary outcome mentions 4 time points, so a repeated measures ANOVA might be more appropriate)

• Thank you for your suggestion. You're right that repeated measures ANOVA may suit the analysis with four time points better.

• Amendment been made in data analysis section by changing the analysis for primary outcome. We will use repeated measure ANOVA since there are 4 time points of data collection (1st, 4th, 8th, and 12th weeks) while secondary outcome remains the same with paired t-test since there are only 2 time point of data collection which is first week (pre) and final week (post).

• See, Data Analysis – 2nd paragraph.

4 For the data analysis section, the authors mention that the data will be checked for normality with appropriate tests. However, there is no mention as to what will happen if the data is found to be not normally distributed. Will data transformations be performed? Will alternate methods be used (Wilcoxon signed rank)? These items should be mentioned in the text as potential alternatives

• We are sorry for missing this point.

• We have mentioned the non-paramettric tests in within the data analysis.

• See, Data Analysis – 3rd paragraph

5 The randomization method also needs attention. As written, it is very unclear how the value of 0 or 1 that gets put into the envelope will be determined. It seems that a sheet with 70 0's and 70 1's (70 sets of numbers) will be created. From there, envelopes will only have one of these numbers. It a simple coin flip method is used (simple randomization), it is unclear how equal sample sizes in each treatment group will be achieved. A more detailed description of the exact method of randomization is needed, and the method used should mention how the end result will be an equal number of subjects in each intervention arm

• Thank you for the constructive input regarding the randomization procedure.

• We have improved the explanation for the procedure for the randomization. I hope this time the explanation is clear and may be replicable to others.

• See, Randomization and allocation concealment – 2nd to 5th sentences.

Table 3. Responses to Reviewer 2

NO COMMENTS RESPONSE

1 The manuscript presents a well-designed protocol for a randomized controlled trial to evaluate the efficacy of a Physiotherapy-led Person-Centered Integrated Care for Older People (PTICOPE) module in enhancing intrinsic capacity among older adults in Malaysia. Here are the key points addressing the review questions:

1. The manuscript provides a valid rationale with clearly identified research questions. The study aims to address the prevalent issue of declining intrinsic capacity in older adults, which impacts their independence and quality of life.

2. The protocol is technically sound and planned to lead to meaningful outcomes. It employs a 12-week, multicenter, randomized controlled trial design with clear intervention and control groups, comprehensive outcome measures, and appropriate statistical analysis methods.

3. The methodology is feasible and described in sufficient detail to allow replication. The protocol outlines participant recruitment, randomization, intervention details, outcome measures, and data analysis plans.

4. The authors have stated that all relevant data from the study will be made available upon study completion.

5. The manuscript is presented in an intelligible fashion and written in Standard English, with a logical structure and clear language.

-Thank you for acknowledging our manuscript as a well-designed protocol as we have worked hard to prepare it. We really appreciate your support and thoughtful feedback and are grateful for the opportunity to contribute to advancing care for older adults in Malaysia.

- We have addressed your suggestion for further improvement diligently and point-by-point as indicated below.

2 Consider providing more details on the PTICOPE module workbook content and how it will be developed based on the WHO-ICOPE framework

• Thank you for the suggestion.

• We have added another paragraph within the PTICOPE development section. The PTICOPE module workbook is designed to align with the WHO Integrated Care for Older People (ICOPE) framework, focusing on key domains of intrinsic capacity, such as mobility, cognition, psychological well-being, vitality, vision, and hearing. The PTICOPE is customized for the Malaysian context, the workbook includes culturally relevant content, evidence-based educational modules, practical exercises, and self-tracking tools. This statement is also supportive of Table 2.

• See, Intervention: Development of PTICOPE – 2nd paragraph

3 Clarify the specific roles of physiotherapists in delivering the intervention and how they will be trained to ensure consistency across centers.

• Thank you for highlighting the importance of clarifying the physiotherapists' roles and their training.

• Before participant recruitment, trained research assistants will be involved, and physiotherapists will deliver the PTICOPE intervention. They will assess intrinsic capacity, guide tailored exercises, provide self-management education, and monitor progress. To ensure consistency, physiotherapists will undergo training on WHO-ICOPE protocols, practical measures, and supervision.

• See, Data collection and monitoring – 1st paragraph

4 Discuss potential limitations of the study and how they will be addressed, such as potential bias in self-reported measures or challenges in long-term follow-up.

• Thank you for highlighting these important issues which we mIght have overlooked.

• These points were discussed in the limitations section as additional factors. Several measures have been proposed for consideration to address these issues effectively.

• See, Limitations and considerations – 4th and 5th paragraphs.

5 Consider including a plan for process evaluation to assess the implementation fidelity and acceptability of the PTICOPE module.

- Thank you for your suggestion to include a plan for evaluation.

• Fidelity and acceptability will be assessed by tracking participants' involvement in the activities using a table where they indicate completion by ticking off each activity.

• See, Limitations and considerations – Last paragraph

6 Provide more information on how the sample size of 70 participants was

determined and whether it is sufficient to detect clinically meaningful differences between groups

• Thank you for your insightful comment on the sample size.

• We have mentioned several parameters, including effect size, to support the calculation of the sample size.

• See, Sample size determination – 1st paragraph.

---

## [Editor Report · Decision Letter 1]

17 Jan 2025

Development and evaluation of a Physiotherapy-led, WHO-ICOPE-Based, Person-Centered Integrated Care Program (PTICOPE) Module to enhance intrinsic capacity in older adults: Protocol for a randomized controlled trial

PONE-D-24-52840R1

Dear Dr. JUSTINE,

We’re pleased to inform you that your manuscript has been judged scientifically suitable for publication and will be formally accepted for publication once it meets all outstanding technical requirements.

Kind regards,

Yogesh Kumar Jain, MPH

Academic Editor

PLOS ONE
---

## [Editor Report · Acceptance letter]

PONE-D-24-52840R1

PLOS ONE

Dear Dr. Justine,

I'm pleased to inform you that your manuscript has been deemed suitable for publication in PLOS ONE. Congratulations! Your manuscript is now being handed over to our production team.

Kind regards,

on behalf of

Dr. Yogesh Kumar Jain

Academic Editor

PLOS ONE